# Dehumanization and mass violence: A study of mental state language in Nazi propaganda (1927–1945)

**Alexander P. Landry** [1] *, **Ram I. Orr** [2], **Kayla Mere** [3]

**1** Organizational Behavior, Stanford Graduate School of Business, Palo Alto, CA, United States of America, **2** Department of Psychology, Tel Aviv University, Tel Aviv, Israel, **3** Psychological and Brain Sciences, University of California, Santa Barbara, Santa Barbara, CA, United States of America

* alandry@stanford.edu

**Data Availability Statement:** Our data is available in the Open Science Framework at https://osf.io/n2kym/.

**Funding:** The author(s) received no specific funding for this work.

## Abstract

Dehumanization is frequently cited as a precursor to mass violence, but quantitative support for this notion is scarce. The present work provides such support by examining the dehumanization of Jews in Nazi propaganda. Our linguistic analysis suggests that Jews were progressively denied the capacity for fundamentally human mental experiences leading up to the Holocaust. Given that the recognition of another's mental experience promotes moral concern, these results are consistent with the theory that dehumanization facilitates violence by *dis*engaging moral concern. However, after the onset of the Holocaust, our results suggest that Jews were attributed a greater capacity for agentic mental states. We speculate this may reflect a process of demonization in which Nazi propagandists portrayed the Jews as highly capable of planning and intentionality while nonetheless possessing a subhuman moral character. These suggestive results paint a nuanced portrait of the temporal dynamics of dehumanization during the Holocaust and provide impetus for further empirical scrutiny of dehumanization in ecologically valid contexts.

## Introduction

Perpetrators of mass violence often appear to deny the humanity of their victims. Theoretical treatments of this topic suggest such *dehumanization* facilitates violence by disengaging moral restraints against harming fellow humans [1]. Although frequently cited in qualitative accounts of ethnic cleansing and genocide (e.g., [2–4]), there remains a dearth of empirical evidence linking dehumanization to such instances of mass violence. This empirical gap has opened the way to suggestions that dehumanization's role in mass violence may be overstated [5, 6]. To evaluate competing perspectives on dehumanization's relationship with mass violence, we examined the dehumanization of Jews in Nazi propaganda both before and during the Holocaust.

### Mind denial and mass violence: Moral disengagement or limited relevance?

Dehumanization can entail the denial of various aspects of a person's humanity, such as personality traits and emotions thought to be unique to humans [7, 8], warmth and competence

**Competing interests:** The authors have declared that no competing interests exist.

[9], or their possession of fundamentally human mental capacities [10] (see [11] for review). Although these forms of dehumanization can be understood as complementary (see [12, 13]), we focus on the denial of fundamentally human mental capacities (i.e., a mind). Considerable research demonstrates that people perceive mind in terms of two primary dimensions: *agency*, the capacity to have complex thoughts and act intentionally, and *experience*, the capacity to feel sensations and emotions [14]. Agency and experience are distinct dimensions of mind but are often moderately correlated [10, 14].

Denying another's mind, particularly their capacity for experience, undermines moral concern for them [10, 15] and has been claimed to facilitate extreme violence [9]. We refer to this claim as the *moral disengagement hypothesis* (see [1]). This may explain why mind denial has been implicated in colonial oppression and other acts of instrumental violence [16, 17]. However, mind denial is also pervasive in everyday life, emerging in the workplace [18], healthcare [19], and men's sexualized perceptions of women [15].

The pervasiveness of mind denial in everyday life lends itself to critiques about whether it indeed facilitates mass violence. Such critiques suggest that perpetrators—rather than deny their victims' minds—often inflict suffering precisely *because* they consider their victims to have malevolent intentions and beliefs [6, 16, 20]. Moreover, that many perpetrators derive sadistic pleasure from degrading their victims suggests they implicitly recognize their victims' capacity to experience this mistreatment [21, 22]. We refer to critiques of mind denial's role in mass violence as the *limited relevance hypothesis*.

## The present research context

Proponents of mind denial's limited relevance to mass violence can also point to the paucity of quantitative evidence linking it—and dehumanization more broadly—to the actual instances of violence it is frequently evoked to explain (e.g., [3, 4]). Therefore, we evaluated these competing perspectives on mind denial's role in mass violence in a historical context cited as a paradigmatic instance of dehumanization by both contemporary survivors and later theorists: the Nazi genocide of European Jewry (e.g., [4, 23]). Of particular relevance to our conceptualization of dehumanization as a process of mind denial, historians of this genocide point to the tendency for perpetrators of all stripes—from high-ranking Nazi officials to grassroots killers—to conceive of their victims as insensate hordes and mindless barbarians [24, 25]. Nonetheless, other work suggests that the Nazis recognized the mental capacities of their Jewish victims because they often portrayed the Jews as insidious and cunning agents of malevolence (e.g., [6]).

**Onset of the Holocaust.** We define the Holocaust as the organized mass murder of individuals on the sole basis of their Jewish identity, and trace its onset to July 1941. During this time, the German occupation forces shifted from relatively circumscribed shootings to the indiscriminate murder of Jewish civilians in Belarus and Lithuania, the systematic execution of Jewish prisoners of war, and the instigation of antisemitic pogroms in occupied Poland and the Baltic states [24, 25, 34].

**A note on perpetrators.** Many accounts of dehumanization's role in mass violence adopt a relatively restricted conception of "perpetrators" as only those who directly engage in killing (e.g., [26]). However, others take a broader perspective that includes ideological architects and mid-level functionaries (e.g., [2]). We employed this broader perspective by identifying as perpetrators the Nazis who produced the propaganda forming the basis of our investigation. Therefore, we assessed a specific form of dehumanization (mind denial) in the rhetoric of a specific subset of perpetrators (ideological architects). Although these genocidal architects likely influenced many other Germans [35] and dehumanized the Jews in other distinct ways

(see Discussion), we refrain from generalizing to other segments of the population or forms of dehumanization.

**Hypotheses.** We tested the moral disengagement and limited relevance hypotheses by analyzing the prevalence of agency and experience mental state terms used when referring to Jews in Nazi propaganda published between 1927 and 1945. If mind denial facilitates violence by undermining moral concern, we would expect a negative trend in mental state terms from 1927 to the onset of the Holocaust. Conversely, if mind denial is of limited relevance to the instigation of mass violence, we would expect no consistent trend in mental state terms preceding the Holocaust. Note that neither of these hypotheses make predictions regarding the trend in dehumanization *after* the onset of mass violence (cf. [26]).

**Exploratory analyses of additional linguistic constructs.** Along with mental state terms, we investigated several other linguistic constructs thought to be of relevance to dehumanization and the historical context of our study.

Concerning dehumanization, we analyzed the expression of *negative emotions* in the propaganda. This was done to determine whether any trends in mental state terms we observed were driven by general negative sentiment, given concerns that dehumanization is merely an expression of negative affect [5, 6]. We also measured language reflecting a desire for social *affiliation* in the propaganda. This was done in order to provide convergent validity for our analysis of experience terms, because a desire for affiliation can motivate people to recognize others' mental experiences [27]. Thus, we reasoned that affiliation terms should show similar patterns as experience terms.

Concerning the historical context—the Nazi regime—we analyzed language related to *health*, *purity*, *death*, and *threat*. We measured health and purity because conceptual metaphors equating Jews to diseases infecting Germany's "national body" (*Volkskörper*) were central to Nazi antisemitism [24, 28] and may have motivated their desire to exterminate the Jews [29, 30]. Indeed, eminent Nazis promised to cleanse the world of a purported "Jewish world plague" [30]. Therefore, we expected such terms would increase leading up to the Holocaust. Likewise, given that Nazi propaganda is thought to have contributed to the Holocaust [25, 31–33], we expected terms related to death to also increase during this time.

The Nazis' conviction that Jews posed an increasing threat may have also contributed to the genocide [25, 33, 34]. Shortly before initiating the Holocaust, Germany launched a grandiose invasion expected to topple the Soviet Union in a matter of weeks [24]. However, the vastly undersupplied *Wehrmacht* soon met drastic setbacks that led even Hitler to question German supremacy [24]). As the wartime tenor in Germany shifted from racist arrogance to existential insecurity, it is conceivable that the Nazis saw the Jews—purported to be masterminding the Soviet war effort abroad and conspiring to "stab Germany in the back" at home [35]—as an increasing threat. This likely contributed to the Nazis' escalation of antisemitic policies to full-fledged genocide in the months following the invasion [25, 33, 34]. Therefore, we expected threat terms to be greater in propaganda published after the onset of the Holocaust (i.e., when Germany's war effort began to disintegrate) than before.

## Methods

### Mind perception dictionary and data collection

We quantified mental state terms with the Mind Perception Dictionary (MPD; [36]), a psycholinguistic tool developed for the Linguistic Inquiry Word Count software (LIWC; [37]). The MPD consists of 326 mental state terms derived from formative studies of mind perception (e.g., [10, 14]), emotional states [38], conceptually related coding schemes [39], and synonyms taken from a standard English dictionary. It classifies each term as either agency (e.g., plan,

think) or experience (e.g., hurt, enjoy), and quantifies the proportion of these terms in a body of text. The MPD reliably captures mental state language in naturalistic contexts [36].

Our data comes from the German Propaganda Archive (https://research.calvin.edu/german-propaganda-archive/), which consists of digitized Nazi propaganda translated to English by a specialist in German propaganda (see [35]). Established in 1999, this voluminous archive now contains hundreds of Nazi-era posters, articles, pamphlets, books, newspapers, and transcripts of political speeches. These materials were intended for a variety of audiences, including children, the general public, and members of the Nazi party. The original German-language versions of the propaganda were collected from the German Federal Archives, university libraries, and the archivist's personal collection used for his own research (see https://research.calvin.edu/german-propaganda-archive/faq.htm for more information on the archive).

As we were interested in the dehumanization of Jews, we established a procedure to only extract text from this archive that directly referred to Jews or could be reasonably inferred to do so. The first author began by reading the first 20 pieces of propaganda in the archive, identifying instances where it was ambiguous as to whether the text was referring to Jews. He then used this experience to draft a coding manual (available at https://osf.io/n2kym/?view_only=514019c7ac3d41f39e8fb35616fb688a) which outlined guidelines for dealing with ambiguous cases (e.g., when to infer terms such as "Bolshevist" or "international finance" were referring to Jews). Using these guidelines, the third author, blind to the research questions at this stage, read each document in the archive and extracted the relevant text. Throughout this process, ambiguous cases were discussed with the first author and data collection guidelines were updated as needed.

To identify trends in agency and experience over time, we organized the text by the month in which it was published. Since the MPD analyzes the proportion of mental state terms in a given body of text, we excluded all months with fewer than 100 total words to avoid inflated estimates. This decision was included in our preregistered analysis plan but was somewhat arbitrary, as there is no consensual minimum of text used for MPD analyses [36]. Ultimately, we aimed to be as inclusive as possible without unduly compromising internal validity. Likewise, although there were other ways we could have segmented the data (e.g., into seasonal or bimonthly components), we chose to segment it by month in order to maximize the number of observations with sufficient text for reliable estimates. Our final dataset thus consisted of 58 months spanning November 1927–April 1945, totaling 57,011 words across 140 individual pieces of propaganda ($M_{\text{word count}}$ = 982.95, $SD$ = 1646.70)—a sufficient amount of text and observations to produce valid estimates using the modeling approach we employed (described in the "Analysis plan" subsection below; [40, 41]).

This data is available at https://osf.io/n2kym/?view_only=514019c7ac3d41f39e8fb35616fb688a and a descriptive breakdown of the different types of propaganda comprising the dataset is provided in S1 Table.

**Control text not referring to Jews.** We sought to evaluate the possibility that our results were driven by general shifts in mental state language in the propaganda over time, rather than changes in reference to Jews specifically. Therefore, after extracting all text referring to Jews to form the primary dataset (hereby "antisemitic text"), the third author returned to each piece of propaganda that was included in this dataset and extracted the same number of words *not* already coded as referring to Jews to form a control text. The exception was cases where over half of the text was already extracted because it referred to Jews ($n$ = 17). In these instances, we simply took all of the remaining text from that piece of propaganda. The control text thus consisted of 51 months spanning from November 1927 to April 1945, totaling 36,391 words across 123 total pieces of propaganda ($M_{\text{word count}}$ = 713.55, $SD$ = 1188.18).

**Psycholinguistic tools for additional linguistic constructs.** We quantified the prevalence of negative emotion (e.g., hate, sad), affiliation (e.g., ally, friend), death (e.g., kill, coffin), and health (e.g., clinic, flu) terms using the default LIWC2015 dictionary [42] and the prevalence of purity terms (e.g., pious, pristine) using the Moral Foundations Dictionary (MFD; [43]), another customized dictionary for LIWC that was developed in a manner similar to the Mind Perception Dictionary. The MFD has undergone extensive psychometric validation and refinement to reliably capture moral sentiments in natural language (e.g., [43–46]). We assessed threat terms (e.g., attack, crisis) with the Threat Dictionary, a psycholinguistic tool containing threat terms derived from word embedding models trained on diverse corpora of text (e.g., Wikipedia entries, Twitter posts; [47]). The Threat Dictionary has shown excellent convergent validity with objective threats and shifts in cultural norms, political attitudes, and macroeconomic activity over the past century [47].

## Analysis plan

**Primary analyses: Mental state terms in antisemitic text.** Our primary analyses were preregistered at https://aspredicted.org/GZC_7X1.

We performed two interrupted time series analyses of (1) agency and (2) experience terms in the antisemitic text. This allowed us to evaluate the moral disengagement and limited relevance hypotheses by examining the trend in these mental state terms leading up to the Holocaust. Moreover, it allowed us to determine whether the onset of the Holocaust influenced mental state terms, and to examine the trend in mental state terms as the Holocaust progressed. We divided the data into a segment of propaganda published before the onset of the Holocaust (pre-onset; all observations before July 1941; $n = 30$) and another segment of propaganda published after its onset (post-onset; all observations from July 1941 onward; $n = 28$). Although there is currently no consensus on the precise date when the Holocaust was initiated [25, 34], "selecting a point of demarcation that generally reflects when the event occurred will still allow the statistical model to assess the impact of the event on the level and trend of the series" ([40]; p. 13).

After segmenting the data, we estimated the pre- and post-onset intercepts and trends in mental state terms with segmented regressions (Eq 1; [40]).

$$y_t = b_0 + b_1 * t + b_2 * event_t + b_3 * t \; after \; event + e_t \tag{1}$$

In this equation, $b_0$ represents the baseline level of mental state terms in November 1927 (the first observation), $b_1$ estimates the trend in mental state terms in the months preceding the Holocaust, and $t$ is a dummy variable denoting time (ranging from 1 to 210 to represent each month from November 1927 to our last observation in April 1945). $b_2$ represents the immediate change in the baseline level of mental states after the onset of the Holocaust and $event_t$ is a dummy variable denoting whether each observation occurred before or after the onset of the Holocaust (0 = pre-onset; 1 = post-onset). $b_3$ estimates the change in trend from pre- to post-onset and $t \; after \; event$ represents how many units after the onset of the Holocaust the observation took place (0 for pre-onset points; 1, 2, 3. . . for subsequent points). $e_t$ represents random variation in the data.

**Additional analyses.** To evaluate the possibility that our primary results were driven by general shifts in mental state language, we repeated these primary analyses on the control text not referring to Jews. We then returned to the antisemitic text. For a more granular perspective on trends in mental states attributed to Jews, we examined changes in specific agency and experience terms from before to after the onset of the Holocaust by constructing Keyness plots [48]. We then examined negative emotions, affiliation, health, purity, and death terms with

additional interrupted time series analyses. Finally, we conducted a two-sample *t*-test to determine whether threat terms were greater in the antisemitic text published after the onset of the Holocaust than before.

Because these analyses followed a set of specified hypotheses based on the constructs' relation to dehumanization (and more specifically, mind denial) and historical accounts of Nazi propaganda, we opted not to adjust our criterion for significance to account for experiment-wide error rate—which would constrain statistical power and may inflate risks of Type-II errors [49, 50].

### Limitations of our data

Of the 210 months from our first to last time point, only 58 had enough text (100 or more words) to render the proportion-based MPD analysis meaningful. The remaining months either had no propaganda published during that time in the archive (*n* = 102) or fewer than 100 total words (*n* = 50). This is problematic because large amounts of missing data can bias model estimates [40]. Moreover, the data was missing not at random, with a substantially larger proportion of data missing in the months preceding the onset of the Holocaust than after its onset (81.25% vs 44%). We encourage readers to keep these limitations in mind when evaluating our results.

## Results

Data and analysis script for the following analyses are available at https://osf.io/n2kym/?view_only=514019c7ac3d41f39e8fb35616fb688a.

### Agency and experience in antisemitic Nazi propaganda

The proportion of agency and experience terms in the dataset were moderately correlated, *r* (58) = .32, *p* = .014, 95% CI [.07, .54], justifying our analysis of them as separate dimensions. Following guidelines from [40], we first converted the proportion of agency and experience mental state terms from each month of propaganda into a time series. We then decomposed the agency and experience series into their trend, seasonal, and irregular components (Fig 1). To isolate any trend in the data, we removed the seasonal component by subtracting it from the raw time series data. We checked for the presence of autocorrelation in the seasonally-adjusted data by performing Ljung-Box tests. There was no autocorrelation in the experience

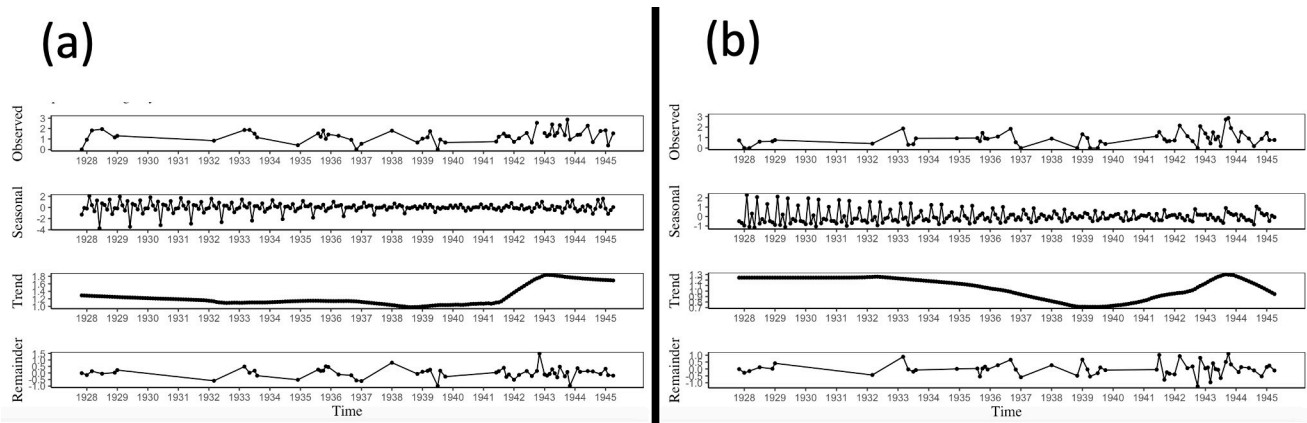

**Fig 1.** Decomposition plots of (a) agency and (b) experience data in antisemitic Nazi propaganda.

data, $\chi2 = 1.85$, $p = .17$, but there was autocorrelation in the agency data, $\chi2 = 10.22$, $p = .02$. Given the presence of autocorrelation in the agency data, we compared the base regression model to an autoregressive model with an AR(1) term, which was determined as the best alternative to the base model upon examination of the ACF plot [40]. Comparing these models' corrected Akaike information criterion (AICc), the base model was a better fit to the agency data: base AICc = 99.78, AR(1) AICc = 102.18. Moreover, a Ljung-Box test on the residuals from the fitted agency data indicated the base model accounted for all autocorrelation, $\chi2 = 0.05$, $p = .82$. Nonetheless, the same pattern of results emerged when using the AR(1) instead of the base model.

We proceeded to fit the segmented regression model to the agency and experience data (Fig 2A and 2B). Before the Holocaust, the trend in agency terms remained relatively constant, $b_1 = -0.002$, 95% CI [-0.01, 0], $SE = 0.002$, $t = -1.21$, $p = .23$, while experience terms decreased significantly, $b_1 = -0.004$, 95% CI [-0.008, -0.001], $SE = 0.002$, $t = -2.21$, $p = .032$. After the onset of the Holocaust, the level of agency terms significantly increased, $b_2 = 0.47$, 95% CI [0.02, 0.91], $SE = 0.22$, $t = 2.21$, $p = .04$, and showed a marginally increasing change in trend from before the Holocaust to after its onset, $b_3 = 0.01$, 95% CI [0, 0.03], $SE = 0.007$, $t = 1.89$, $p = .064$. The level of experience terms, on the other hand, did not significantly differ from pre-to-post onset, $b_2 = 0.32$, 95% CI [-0.20, 0.84], $SE = 0.26$, $t = 1.26$, $p = .22$, nor was there a significant change in trend, $b_3 = 0.009$, 95% CI [0, 0.02], $SE = 0.008$, $t = 1.18$, $p = .24$.

## Agency and experience in control text

We next performed interrupted time series analyses of agency and experience terms in the control text that did not refer to Jews. The control text consisted of 51 observations, as compared to the 58 comprising the antisemitic text (see "Mind perception dictionary and data

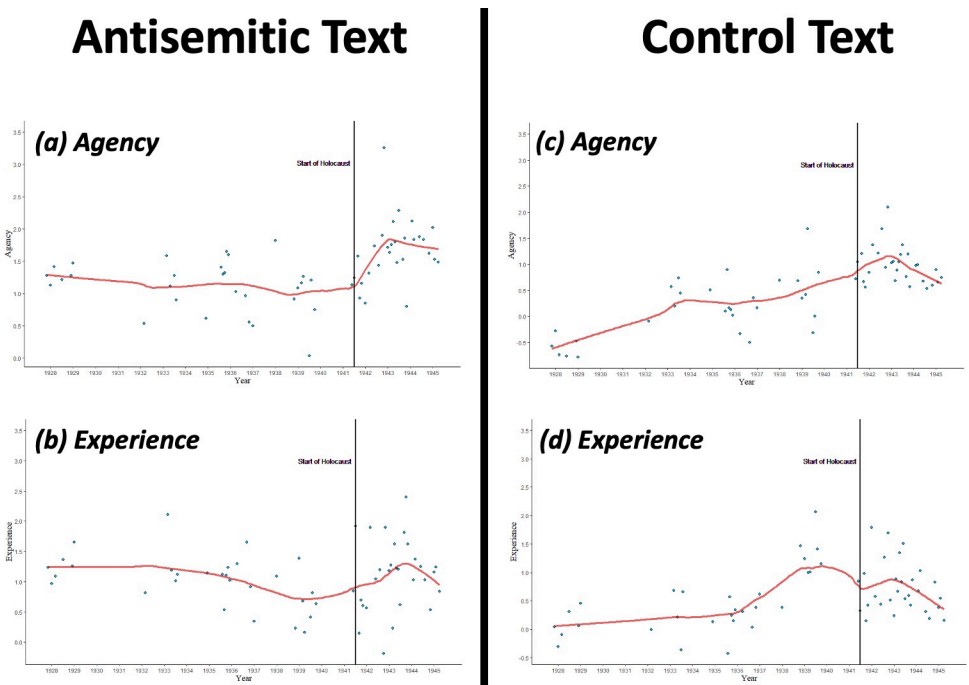

**Fig 2. Interrupted time series analyses of agency and experience in Nazi propaganda.** Interrupted time series plots for (a) agency and (b) experience terms in the antisemitic text and (c) agency and (d) experience terms in the control text.

collection" section). While this difference may seem slight, some cycle subseries in the control text only had a single observation and as such the decomposition algorithm was unable to fit a local linear model to all of the subseries. Therefore, we imputed the seven missing values in this dataset with a linear interpolation based on a robust STL decomposition method: the "na. interp" function available through the *forecast* package for R software (Version 8.15; [51]).

As with the antisemitic text, we decomposed and seasonally adjusted the agency and experience data. We checked for the presence of autocorrelation in the seasonally adjusted data by performing Ljung-Box tests. There was autocorrelation in both the agency and experience data (agency: $\chi2$ = 13.05, $p$ < .001; experience: $\chi2$ = 4.33, $p$ = .037). Therefore, as with the antisemitic text, we compared the base models to autoregressive models with an AR(1) term. In both cases, the base model was a better fit (agency: base AICc = 105.18, AR(1) AICc = 105.97; experience: base AICc = 93.07, AR(1) AICc = 94.31). Moreover, a Ljung-Box test on the residuals from the fitted data indicated the base model accounted for all autocorrelation (agency: $\chi2$ = 0.39, $p$ = .53; experience: $\chi2$ = 0.64, $p$ = .42). Nonetheless, for both the agency and experience data, the same pattern of results emerged when using the AR(1) instead of the base model.

We then fit segmented regression models to the data (Fig 2C and 2D). Whereas agency terms in the antisemitic text showed no significant trend before the Holocaust, agency terms in the control text showed an increasing trend during this time, $b_1$ = 0.01, 95% CI [0.005, 0.011], $SE$ = 0.001, $t$ = 4.92, $p$ < .001. Likewise, whereas the trend in experience terms in the antisemitic text decreased before the Holocaust, the trend in experience terms in the control text *increased* during this time, $b_1$ = 0.01, 95% CI [0.004, 0.011], $SE$ = 0.002, $t$ = 4.55, $p$ < .001. However, after the onset of the Holocaust, the level of agency terms in the control text increased, $b_2$ = 0.45, 95% CI [0.03, 0.87], $SE$ = 0.21, $t$ = 2.15, $p$ = .036, and the level of experience terms did not change, $b_2$ = -0.20, 95% CI [-0.67, 0.27], $SE$ = 0.23, $t$ = -0.86, $p$ = .40—both of which also occurred in the antisemitic text. Nonetheless, whereas agency terms in the antisemitic text showed an increasing change in trend from before the Holocaust to after its onset, the change in trend for agency terms in the control text *decreased* over this period, $b_3$ = -0.02, 95% CI [-0.03, -0.001], $SE$ = 0.01, $t$ = -2.75, $p$ = .008. Likewise, whereas experience terms in the antisemitic text did not show a significant change in trend from pre- to post-onset, experience terms in the control text showed a decreasing change over this period, $b_3$ = -0.02, 95% CI [-0.03, -0.002], $SE$ = 0.01, $t$ = -2.23, $p$ = .03. These generally divergent results suggest that the findings from the antisemitic text were not driven by general shifts in mental state language over time.

## Keyness plots of agency and experience in antisemitic text

We constructed Keyness plots to visualize how specific agency and experience terms differed in prevalence from before to after the onset of the Holocaust (Fig 3). These plots were derived using Chi-square tests that compared the relative frequency of a word in one subset of the text against the expected frequency based on the other subset (e.g., the experience term "hatred" appeared much more frequently in the post-onset subset of text than would be expected given its frequency in the pre-onset text, while "honor" was much less so; [48, 52]). Examining agency terms (Fig 3A), those referring to intentionality (e.g., plans, goal) and malevolence (infernal, evil) increased following the onset of the Holocaust, while those referring to benevolence (love, benevolent) decreased. Likewise, several malevolent experience terms (e.g., sadistic, hatred) increased following the onset of the Holocaust, while some benevolent experience terms (honor, concerned) decreased (Fig 3B). We speculate on these patterns in the Discussion.

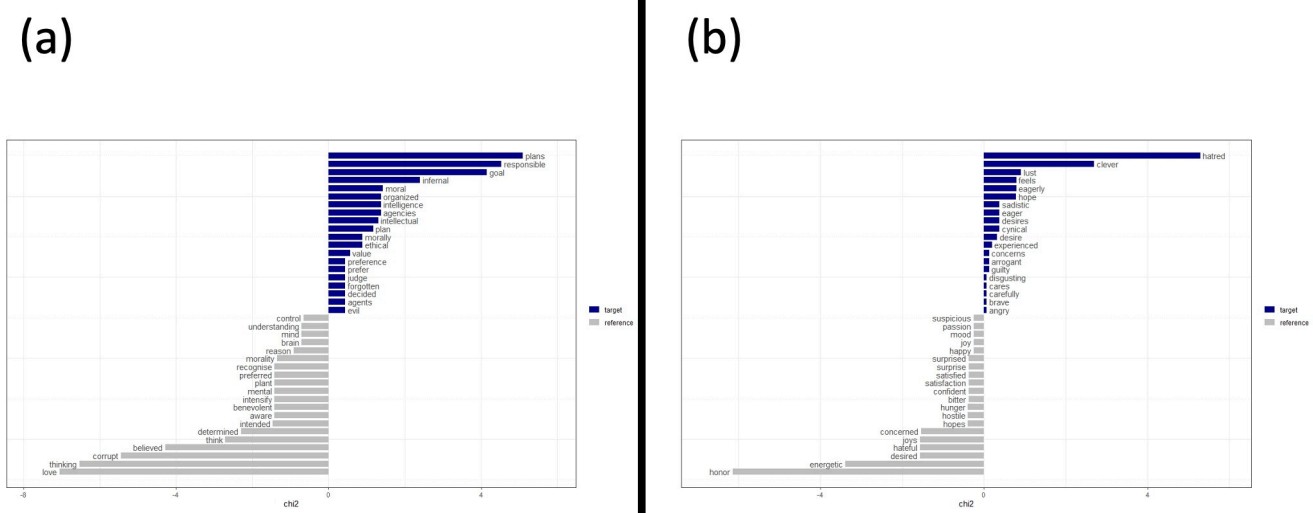

**Fig 3.** Keyness plots depicting changes in (a) agency and (b) experience terms in antisemitic propaganda from pre- to post-onset of the Holocaust.

### Additional linguistic constructs

Below we provide a narrative overview of the results concerning the additional linguistic constructs. The full reporting of these results can be found in the "Additional linguistic constructs: detailed results" section of the S1 File.

We first tested whether the decline in experience terms in the antisemitic text preceding the Holocaust was driven by general negative sentiment. We found no evidence for this possibility, as there was no trend in negative emotions preceding the Holocaust. We then sought convergent support for the finding that experience terms decreased preceding the Holocaust by analyzing affiliation terms, because a desire for affiliation motivates people to recognize others' mental experiences [27]. Consistent with the declining trend in experience, we observed a marginally significant decline in affiliation terms prior to the onset of the Holocaust.

We also observed increasing trends in health and purity terms leading up to the Holocaust —consistent with the notion that rhetoric likening Jews to infectious diseases [24, 28] motivated the Nazis' desire to purify Germany's "national body" (*Volkskörper*) by exterminating them [24, 28–30]. Death terms also increased preceding the Holocaust, supporting the notion that Nazi propaganda played a role in inciting the genocide [25, 31–33]. Finally, we found that antisemitic propaganda published after the onset of the Holocaust contained a greater proportion of threat terms than that published before the Holocaust. This accords with the possibility that the Nazis ramped up their genocidal policies because they perceived the Jews as an increasing threat after the war effort began to decline [25, 34].

We note that the magnitude of several statistically significant effects were small, and elaborate on this in the Discussion.

### Discussion

Dehumanization is often evoked to explain instances of mass violence. Proponents of this view claim that the denial of their victims' humanity enables perpetrators to overcome moral inhibitions against harming conspecifics (e.g., [1, 3, 4]). However, empirical evidence linking dehumanization to actual instances of mass violence is lacking. Moreover, a growing body of scholars have questioned this "moral disengagement" hypothesis by pointing out that

perpetrators often inflict harm precisely because they recognize their victims' capacity for fundamentally human mental states—concluding that dehumanization is of limited relevance to mass violence (e.g., [6, 16, 20–22]).

To evaluate these competing perspectives, we examined the prevalence of fundamentally human mental state terms in Nazi antisemitic propaganda. Terms reflecting the capacity to experience sensations and emotions steadily decreased leading up to the Holocaust. Because recognizing another's mental experiences promotes moral concern toward them [10], this progressive denial of experience preceding the Holocaust accords with the notion that dehumanization facilitates violence by disengaging moral restraints (e.g., [3]). However, after the onset of the Holocaust, we observed an *increase* in agentic mental state terms—specifically terms related to intentionality and malevolence (Fig 3A). Several experience terms referring to malevolence also increased following the onset of the Holocaust, as did the term "guilty"—with its clear implication of moral culpability (Fig 3B). These patterns are consistent with claims that the Jews were subject to *demonization*, in which their capacity for sophisticated reasoning was perceived to coexist with a subhuman moral depravity [2, 4] (see also [53, 54]). For instance, [55] describes groups perceived to be simultaneously agentic and subhuman as "...out-and-out demons—groups that manage to be both repulsively subhuman and despicably evil. This is how the Nazis saw the Jews" (p. 328).

In other words, although agency increased in antisemitic propaganda after the onset of the Holocaust, the Jews may have been dehumanized in other ways during this time. Indeed, although our conception of dehumanization as mind denial offered us a precise conceptualization of the construct [56] and a well-validated empirical tool to operationalize it (the MPD; [36]), dehumanization can take many forms other than mind denial [57]. Consistent with the possibility that the Nazis attributed the Jews agency while also ascribing them repulsively subhuman qualities, Nazi propaganda frequently equated Jews to disgusting animals and diseases [58]. Therefore, we encourage future research to examine dehumanization in genocidal contexts with broader operationalizations of the construct (see [11]).

## Speculations on increasing agency and additional linguistic constructs

Nazi propogandists' demonization of the Jews following the onset of the Holocaust may reflect a shift toward offering a palliative rationalization for the many executioners who experienced revulsion and trauma after shooting Jewish civilians [59]. For instance, given the close link between attributions of agency and moral responsibility [60], portraying the Jews as intentionally malevolent may reflect the "victim blaming" thought to enable the perpetration of inhumanities [1].

The disintegration of the Nazi war effort following their ill-fated invasion of the Soviet Union may also have catalyzed the attribution of agency to Jews [24] (see also [61]). As the wartime tenor in Germany shifted from racist arrogance to existential insecurity, it is conceivable that the Nazis saw the Jews—believed to be masterminding the Soviet war effort abroad while conspiring to "stab Germany in the back" at home [35]—as a particularly insidious threat. Indeed, threat terms were greater in antisemitic propaganda published after the onset of the Holocaust than before. When facing members of a threatening outgroup, individuals are motivated to consider their intentions and so attribute them greater agency [27, 62], which may have been reflected in the wartime propaganda of this period.

Three further findings bolster our confidence that Jews were dehumanized through the progressive denial of mental experiences leading up to the Holocaust. First, given concerns that instances of dehumanization instead merely reflect general negative sentiment toward the outgroup (e.g., [6]), it is important to note that the decline in experience terms preceding the

Holocaust did not appear to be driven by general negative sentiment in the propaganda text. Second, experience terms did *not* show a similar decline in the control text—suggesting that the effect was not due to general shifts in language. Third, we observed a similar decrease in affiliation terms preceding the Holocaust, which we expected because a desire for affiliation motivates people to recognize others' mental experiences [27].

We also provide support for rich qualitative accounts by historians, linguistics, social psychologists, and sociologists of several factors thought to contribute to the Holocaust. For instance, Nazi propaganda likening Jews to infectious diseases is thought to have motivated support for their attempted extermination [24, 28–30]. Indeed, terms related to health and purity increased in Nazi propaganda preceding the Holocaust (see also [58]). We also bolster claims that Nazi propaganda contributed to the Holocaust by the finding that death terms in such propaganda steadily increased leading up to its onset [25, 31–33]. Finally, our previously discussed finding that threat terms increased in Nazi propaganda suggests that their perception of the Jews as a dire threat may have facilitated the escalation to genocide [25, 33, 34].

## Limitations

As noted previously, a crucial limitation of the present research is the large proportion of missing observations in our dataset. Although time series analyses are certainly feasible in the presence of missing data [63], large amounts of missing values can bias model estimates [40]. This problem was compounded by the fact that the data appeared to be missing not at random, with a larger proportion of values missing in the subset of data published before the onset of the Holocaust. Indeed, although the German Propaganda Archive is impressive in the volume and diversity of material it contains, it should in no way be considered representative of propaganda produced under the Nazi regime. Moreover, we used only a single coder to extract the data from the achieve, which prevented us from determining the reliability of the extraction procedure. Therefore, we consider our results suggestive rather than conclusive and encourage future research to investigate more comprehensive corpora of Nazi propaganda.

It should also be noted that many of the significant effects we observed were small in magnitude and some may have been spurious due to a lack of accounting for experiment-wise error. The small magnitude of these effects is understandable, however, as the portrayal of Jews in Nazi propaganda was influenced by a multitude of factors other than temporal distance from the Holocaust—including the codification of antisemitic legislature in Germany, the economic collapse brought on by the Great Depression, and the rise of anti-Nazi movements thought to be orchestrated by Jewish conspirators [35]. A perhaps more serious limitation of our work lies in our method of quantifying mental states. Although the MPD is a well-validated measure of mental state language, it relies on a relatively simplistic word count method that likely omits important nuance in dehumanizing rhetoric. Therefore, scholars could augment our "blunt force" word count approach with more sophisticated natural language processing (see [64]). More generally, we encourage future research to leverage recent advances in the computational social sciences toward ecologically valid studies of dehumanization (see [11]).

## Conclusion

Mental state terms related to experience declined in Nazi antisemitic propaganda leading up to the Holocaust, consistent with the notion that dehumanization undermines moral restraints against violence [3]. Following the onset of the Holocaust, however, agentic mental state terms increased. This may have reflected the propagandists' demonization of the Jews in order to provide a palliative rationalization for their mass murder, and/or focus the population on the

Jews' intentions and plans to counter the dire threat they were perceived to pose. Although the limitations of our data render our results suggestive, such possibilities should impel further empirical scrutiny of dehumanization in genocidal contexts. Although a daunting task, we maintain that psychological science can offer crucial insights into the human proclivity for mass violence and hope the present work promotes future inquiry.

## Supporting information

**S1 File. Supporting information.**
(DOCX)

**S1 Table. Information on the propaganda comprising our corpus.** [a] Internal sources were those produced for Nazi party members (e.g., bulletins instructing Nazi leaders on how to compose their speeches). [b] External sources were those produced for a public audience (e.g., widely disseminated posters).
(DOCX)

## Acknowledgments

We thank Joshua Jackson for his gracious feedback, Andrew Jebb for his statistical advice, and Adam Waytz for his guidance throughout the duration of this project.

## Author Contributions

**Conceptualization:** Alexander P. Landry.

**Data curation:** Kayla Mere.

**Formal analysis:** Ram I. Orr.

**Investigation:** Kayla Mere.

**Methodology:** Alexander P. Landry, Ram I. Orr.

**Visualization:** Ram I. Orr.

**Writing – original draft:** Alexander P. Landry, Ram I. Orr.

**Writing – review & editing:** Alexander P. Landry, Ram I. Orr.

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
