## [Decision Letter · Decision Letter 0]

30 Jun 2022

PONE-D-21-38261

Dehumanization and mass violence: A study of mental state language in Nazi propaganda (1927-1945)

PLOS ONE

Dear Dr. Landry,

Thank you for submitting your manuscript to PLOS ONE. After careful consideration, we feel that it has merit but does not fully meet PLOS ONE’s publication criteria as it currently stands. Therefore, we invite you to submit a revised version of the manuscript that addresses the points raised during the review process.

The reviewers have raised concerns about the motivation and utilization of the proposed work, the lack of detailed methodological description, lack of depth in evaluation and results, lack of sufficient details on data and data collection, missing data, and poor quality figures.  Please address these concerns considering the PLOS ONE’s publication criteria.

We look forward to receiving your revised manuscript.

Kind regards,

Rashid Mehmood, PhD

Academic Editor

PLOS ONE

Journal Requirements:

Reviewers' comments:

Reviewer's Responses to Questions

**Comments to the Author**

1. Is the manuscript technically sound, and do the data support the conclusions?

Reviewer #1: No

Reviewer #2: No

2. Has the statistical analysis been performed appropriately and rigorously? 

Reviewer #1: No

Reviewer #2: No

3. Have the authors made all data underlying the findings in their manuscript fully available?

Reviewer #1: Yes

Reviewer #2: No

4. Is the manuscript presented in an intelligible fashion and written in standard English?

Reviewer #1: Yes

Reviewer #2: No

5. Review Comments to the Author

Reviewer #1: In the notion that recognition of another's mental experience encourages moral concern, dehumanization facilitates violence because it disengages moral concern. In this study, Landry et al. provided evidence of the relation between dehumanization and mass violence, by examining the dehumanization of Jews in Nazi propaganda. Dehumanization can refer to the denial of a person's humanity in several ways; for example, their human characteristics and emotions, their warmth and competence, or their ability to possess fundamentally human mental faculties. Landry et al. focused only on the denial of fundamentally human mental capacities. Before the time of the Holocaust, they found from the linguistic analysis of Nazi propaganda that Jews were progressively denied the capacity for fundamentally human mental experiences.

In light of the Holocaust, their results suggest that the Jews were perceived as having a higher capacity for agentic states of mind.

This may be due to a process of dehumanization during which Nazi propagandists portrayed the Jews as highly intelligent and intentional, but subhuman morally. They claimed many hypotheses: the moral disengagement hypothesis, the limited relevance hypothesis. They tested it by analyzing the prevalence of agency and experience mental state terms used when referring to Jews in Nazi propaganda from 1927 to 1945. The main tests consisted of two interrupted time series analyses using R focusing on those terms. They divided the data into a segment of propaganda published before the onset of the Holocaust and another segment of propaganda published after its onset. They estimated the pre-and post-event intercepts and trends in mental state terms with segmented regressions. In Nazi antisemitic propaganda leading up to the Holocaust, terms related to experience declined, consistent with the notion that dehumanization weakens moral restraints against violence. During the Holocaust, propaganda displayed an increase in agentic mental state terms. In order to provide a palliative rationalization for their mass murder or (2) provide a palliative explanation for their dehumanization, propagandists may have demonized the Jews. Concentrate the population's attention on the intentions and plans of the purported Jewish Bolsheviks.

The paper is well written and follows a standard rational development, methodology, and analysis. The authors need to improve the presentation and discussion of the results.

Reviewer #2: The authors provided quantitative analysis for the notion that the dehumanization of human beings leads to mass violence. The purpose of their work is to examine the temporal dynamics of dehumanizing Jewish people in Nazi propaganda between 1927 and 1945. The dataset consisted of 58 months spanning from November 1927 to April 1945, totaling 57,011 words across 140 individual pieces of propaganda. The results shows that Jews had a greater capacity for agentic mental states.

Comments:

- The authors should explain what is the use of this work today?

- Discuss results in more details.

- The authors need to describe the methodology in more details.

- The authors need to provide more details regarding data collection

- The authors must provide better quality figures.

6. PLOS authors have the option to publish the peer review history of their article (what does this mean?). If published, this will include your full peer review and any attached files.

Reviewer #1: No

Reviewer #2: No

---

## [Author Response · Author response to Decision Letter 0]

27 Jul 2022

Below I will respond to the reviewer and editor comments raised in the decision letter. However, I also wrote this response in a Word document included in the revised submission package, which may be easier to read as it included clearer subheadings and hyperlinked text. 

To whom it may concern:

My colleagues and I would first like to sincerely thank the editor and four anonymous reviewers for their careful consideration of the initial draft our manuscript, entitled “Dehumanization and mass violence: A study of mental state language in Nazi propaganda (1927-1945)” (Manuscript ID: PONE-D-21-38261). We respectfully request your reconsideration. 

We received gracious feedback from the reviewers on July 20, 2022. In total, they offered 14 distinct actionable suggestions to improve our manuscript. We took each piece of advice into careful consideration, and I hope this is reflected in an improved manuscript. In the following, I will respond to each distinct point raised by the reviewers, noting our revisions, in the order they appeared in the original decision letter. 

Firstly, we note that Reviewers 1 and 2 each made several general suggestions that we believe we address in our revisions based on the more targeted suggestions of Reviewers 3 and 4. Namely, Reviewer 1 asked us to “improve the presentation and discussion of the results”, while Reviewer 2 suggested we discuss the procedure for data collection, methodology, and results in more detail.

Contemporary relevance of our research. Given that our study was focused on a specific historical context (Nazi propaganda published from 1927-1945) Reviewer #2 asked us to elaborate on the relevance of our work today. We first note that throughout the introduction and conclusion, we place theoretical emphasis on a current tension in the dehumanization literature of interest to many contemporary social psychologists—that is, differing perspectives on dehumanization’s role in mass violence. As discussed in the main text (see especially the subsection of the introduction titled “Mind denial and mass violence: moral disengagement or limited relevance?”; lines 56-87), it was long assumed that dehumanization facilitates violence by disengaging moral restraints against harming conspecifics (e.g., Kelman, 1973). However, recent prominent critiques have questioned this hypothesis on both conceptual and empirical grounds (e.g., Bloom, 2017; Enock, 2022; Lang, 2020; Over, 2021). Our study offers an ecologically valid test of these competing perspectives that we believe to be of interest for contemporary social psychologists, and say as much in our Discussion (lines 566-580).

Moreover, we discuss how this work has implications for future research. Namely, we suggest that our study lays a foundation for further empirical scrutiny of dehumanization—and other social-psychological constructs—in ecologically valid contexts, particularly by leveraging recent advances in computational social science (e.g., Mendelsohn et al., 2020; see Abstract and lines 708-711, 718-722).

Revised Figure 2. Reviewer #2 noted the low quality of our figures. In light of this, we revised Figure 2. To more clearly show the dispersion of data and trend line, we shortened the Y-axis and made the datapoints and smoothed regression lines smaller. We also altered the color palette in an effort to make the figure more aesthetically pleasing. We believe Figures 1 and 3 are of sufficient quality and therefore did not revise them. 

Describing the German Propaganda Achieve. Reviewer 3 asked us to provide more deail on our source of Nazi propaganda, so we now expand on our description of the German Propaganda Achieve: “Established in 1999, this voluminous archive now contains hundreds of Nazi-era posters, articles, pamphlets, books, newspapers, and transcripts of political speeches (see “Examples of Nazi propaganda” in Supporting Information). These materials were intended for a variety of audiences, including children, the general public, and members of the Nazi party. The original German-language versions of the propaganda were collected from the German Federal Archives, university libraries, and the archivist’s personal collection used for his own research (see https://research.calvin.edu/german-propaganda-archive/faq.htm for more information on the archive)” (lines 191-199).

Elaborating on data extraction. Reviewers 1 and 3 each asked us to provide more details on how the control text was extracted from the propaganda. We have now provided additional detail on the development of the control text: “After extracting all text referring to Jews to form the primary dataset (hereby ‘antisemitic text’), the third author returned to each piece of propaganda that was included in this dataset and extracted the same number of words not already coded as referring to Jews to form a control text. The exception was cases where over half of the text was already extracted because it referred to Jews (n = 17). In these instances, we simply took all of the remaining text from that piece of propaganda” (lines 239-244).

Justifying our level of analysis. Reviewer 3 asked why we chose the month as our level of analysis (rather than, for instance, dividing the data into bimonthly or seasonal components). We decided to analyze the data at the level of the month in order to be as inclusive as possible without unduly compromising internal validity. This was included in our preregistered analysis plan: https://aspredicted.org/GZC_7X1. Nonetheless, we readily accept that there are alternative ways of analyzing these data and provide both the raw data and analysis script on an open-access repository should others wish to perform these alternative analyses: https://osf.io/n2kym/?view_only=514019c7ac3d41f39e8fb35616fb688a

Keyness plot. Reviewer 3 noted that the methodology and results from the Keyness plots we constructed could have been elaborated on. We do so now in the “Keyness plots of agency and experience in antisemitic text” subsection of the Results (lines 455-492). For one, we elaborate on the methodology used to construct the plots and provide supporting references for a more comprehensive overview: “These plots were derived using Chi-square tests that compared the relative frequency of a word in one subset of the text against the expected frequency based on the other subset (e.g., the experience term “hatred” appeared much more frequently in the post-onset subset of text than would be expected given its frequency in the pre-onset text, while “honor” was much less so; Scott, 1997; Scott & Tribble, 2006)” (lines 457-461). We also expand on the results in this section, “Examining agency terms (Fig 3a), those referring to intentionality (e.g., plans, goal) and malevolence (infernal, evil) increased following the onset of the Holocaust, while those referring to benevolence (love, benevolent) decreased. Likewise, several malevolent experience terms (e.g., sadistic, hatred) increased following the onset of the Holocaust, while some benevolent experience terms (honor, concerned) decreased (Fig 3b)” (lines 462-491). We elaborate on these results in the discussion when speculating that the Jews were subject to demonization following the onset of the Holocaust: “after the onset of the Holocaust, we observed an increase in agentic mental state terms—specifically terms related to intentionality and malevolence (Fig 3a). Several experience terms referring to malevolence also increased following the onset of the Holocaust, as did the term “guilty”—with its clear implication of moral culpability (Fig 3b). These patterns are consistent with claims that the Jews were subject to demonization, in which their capacity for sophisticated reasoning was perceived to coexist with a subhuman moral depravity [2,4] (see also [53,54])” (lines 580-600). 

Agency and experience correlation. In the Introduction of our original manuscript, we support the claim that “Agency and experience are distinct dimensions of mind but are often moderately correlated” by presenting the correlation (r = .32) from our own data. Reviewer 4 suggested we instead move this empirical result to the Results section and support the claim with another reference. We have done so, supporting our assertion by referencing seminal papers on mind perception (Waytz et al., 2010; Gray et al., 2007; line 66) and moved the reporting of the correlation between the proportion of agency and experience terms in our own data to the Results section—including the accompanying N and confidence interval (lines 348-349).

Restructuring our presentation of the additional linguistic constructs. Along with our focal construct of mental state terms, we also analyzed several additional linguistic constructs that were theoretically relevant to either dehumanization or the historical context in which we were studying it—the Nazi regime. However, Reviewers 3 and 4 both noted that the introduction, operationalization, reporting of results, and discussion of these additional analyses and constructs was disorganized and difficult to follow. Upon further review, we concur and have thus substantially reorganized the presentation of these constructs. 

In the introduction, we now clearly introduce each of these additional constructs in a subsection titled “Additional linguistic constructs” (lines 124-155). This subsection provides the theoretical rationale for investigating each of the six constructs we examined, on the basis of critiques and theories of dehumanization (for negative emotion and affiliation terms; lines 127-133) or historical accounts of the Nazi regime (for health, purity, death, and threat terms; lines 134-155). 

We clarify the operationalization of these additional constructs in the subsection of the Method section titled “Psycholinguistic tools for additional linguistic constructs” (lines 247-271), describing the psycholinguistic tools used to operationalize negative emotion, affiliation, health, and death terms (the default LIWC2015 dictionary; Pennebaker et al., 2015), purity terms (the Moral Foundations Dictionary for LIWC; Graham et al., 2009) and threat terms (the Threat Dictionary; Choi et al., 2022). We also provide an overview of the analytic procedure for each of these linguistic constructs in the “Additional analyses” subsection of the Analysis Plan (lines 317-331). 

We provide a narrative overview of the results of these analyses in the main text (lines 495-565). As in the introduction, we structure this section by organizing the results for each construct in terms of the theoretical rational for analyzing it (i.e., results related to dehumanization theory and those related to factors thought to contribute to the Holocaust. Along with this narrative overview, we provide detailed reporting of the results in the Supporting Information (lines 2-56). 

Finally, we have created a new subsection in the Discussion titled “Additional linguistic constructs” (lines 662-683) in which we elaborate on the implications of these results for our findings regarding mental states and dehumanization theory (lines 663-671) and how they bolster accounts of factors thought to have facilitated the Nazi Holocaust (lines 672-683). 

Contextualizing our dataset. Although Reviewer 4 noted that the size of our dataset (approximately 57,000 words spanning 58 months) “seems sufficiently sizable for statistical analysis and drawing certain logical inferences”, they advised us to make the sufficient size of our dataset more explicit by referencing methodological discussions of sample size for linguistic analyses. We have now done so when introducing our data in the Method section, saying “The number of text and observations in this dataset produce valid estimates using the modeling approach we employed” and providing supporting references (Jebb et al., 2015; McCleary et al., 1980; lines 224-226).

Justification for foregoing correction for experiment-wise error. As per Reviewer 4’s suggestion, we now provide justification for foregoing correction for experiment-wise error for the multiple tests performed when analyzing the additional linguistic constructs in the “Analysis plan” subsection of the Method (lines 327-331): “Because these analyses followed a set of specified hypotheses based on the constructs’ relation to dehumanization (and more specifically, mind denial) and historical accounts of Nazi propaganda, we opted not to adjust our criterion for significance to account for experiment-wide error rate—which would constrain statistical power and may inflate risks of Type-II errors (Bender & Lange, 2001; Perneger, 1998).” 

Confidence intervals and effect size. As per Reviewer 4’s suggestions, we have now added confidence intervals for all results reported in the main text and Supporting Information, as well as a measure of effect size (Cohen’s d) for the two-sample t-test reported in the Supporting Information (line 55).

Noting small effect sizes. Reviewer 4 suggested we draw more attention to the magnitude of our observed effects in order to help readers appraise our conclusions. We wholeheartedly agree with this suggestion. Indeed, in our Discussion, we explicitly draw attention to this issue, noting that “many of the effects we observed were small in magnitude” (line 695) and discussing several possibilities for this. Moreover, we now explicitly “note that the magnitudes of several statistically significant effects were small” in the Results section as well (line 564). 

Discussion. Reviewer 4 suggested that the start of the Discussion would be made stronger and clearer if explicitly tied back to hypotheses, so we have now done so (lines 567-574): “Dehumanization is often evoked to explain instances of mass violence. Proponents of this view claim that the denial of their victims’ humanity enables perpetrators to overcome moral inhibitions against harming conspecifics (e.g., [1,3,4]). However, empirical evidence linking dehumanization to actual instances of mass violence is lacking. Moreover, a growing body of scholars have questioned this “moral disengagement” hypothesis by pointing out that perpetrators often inflict harm precisely because they recognize their victims’ capacity for fundamentally human mental states—concluding that dehumanization is of limited relevance to mass violence (e.g., [6,16,20–22]).” 

Limitations of our data. Reviewer 4 noted an important limitation to the present research that we did not draw attention to in our initial version of the manuscript. Namely, they pointed out that there is “Inherent uncertainty about the comprehensiveness of the sampling source.” Therefore, we now note that “although the German Propaganda Archive is impressive in the volume and diversity of material it contains, it should in no way be considered representative of propaganda produced under the Nazi regime. Therefore, we consider our results suggestive rather than conclusive and encourage future research to investigate more comprehensive corpora of Nazi propaganda“ in the “Limitations” subsection of our Discussion (lines 690-694). 

Concluding remarks. We again thank the editor and reviewers for their careful consideration of our manuscript. We did our best to integrate this feedback, and would genuinely appreciate your reconsideration of what is hopefully a substantially improved paper.

Sincerely, 

Alexander Landry

Doctoral Student, Organizational Behavior

Stanford University

---

## [Decision Letter · Decision Letter 1]

22 Aug 2022

PONE-D-21-38261R1Dehumanization and mass violence: A study of mental state language in Nazi propaganda (1927-1945)PLOS ONE

Dear Dr. Landry,

Thank you for submitting your manuscript to PLOS ONE. After careful consideration, we feel that it has merit but does not fully meet PLOS ONE’s publication criteria as it currently stands. Therefore, we invite you to submit a revised version of the manuscript that addresses the points raised during the review process. Please address Reviewer 4 comments.

We look forward to receiving your revised manuscript.

Kind regards,

Rashid Mehmood, PhD

Academic Editor

PLOS ONE

Journal Requirements:

Reviewers' comments:

Reviewer's Responses to Questions

**Comments to the Author**

1. If the authors have adequately addressed your comments raised in a previous round of review and you feel that this manuscript is now acceptable for publication, you may indicate that here to bypass the “Comments to the Author” section, enter your conflict of interest statement in the “Confidential to Editor” section, and submit your "Accept" recommendation.

Reviewer #1: All comments have been addressed

Reviewer #2: All comments have been addressed

Reviewer #3: (No Response)

Reviewer #4: (No Response)

2. Is the manuscript technically sound, and do the data support the conclusions?

Reviewer #1: Yes

Reviewer #2: Yes

Reviewer #3: Yes

Reviewer #4: Yes

3. Has the statistical analysis been performed appropriately and rigorously? 

Reviewer #1: Yes

Reviewer #2: Yes

Reviewer #3: Yes

Reviewer #4: Yes

4. Have the authors made all data underlying the findings in their manuscript fully available?

Reviewer #1: Yes

Reviewer #2: Yes

Reviewer #3: Yes

Reviewer #4: Yes

5. Is the manuscript presented in an intelligible fashion and written in standard English?

Reviewer #1: Yes

Reviewer #2: Yes

Reviewer #3: Yes

Reviewer #4: Yes

6. Review Comments to the Author

Reviewer #1: The authors have addressed my recommendations. There is an improvement in the presentation and discussion of the results.

Reviewer #2: The authors made fine progress except that Figure 1 and Figure 3 still have poor quality. The authors should provide better quality figures.

Reviewer #3: The authors have been responsive to my six original concerns. The manuscript reads better and is more compelling and well structured as a result. My only small quibble is with the response to my point 3. The authors state that "We decided to analyze the data at the level of the month in order to be as inclusive as possible without unduly compromising internal validity." It seems to me that taking one month as the temporal unit of analysis actually reduces inclusiveness because it excludes some data that appears in sparse months (fewer than 100 words) from the analysis, as well as excluding some months (albeit empty ones) from the overall coverage. It's hard to see how 'internal validity' would be compromised by a somewhat coarser unit (e.g., two-month) given that the preferred unit is rather arbitrary and the analysis doesn't stand or fall on very fine-grained data patterns. But I do accept that the one-month unit is defensible and that others can re-analyze the data as they see fit, so I don't look on this as an obstacle to publication.

Reviewer #4: PONE-D-21-38261_R1

This revised manuscript, Dehumanization and mass violence: A study of mental state language in Nazi propaganda (1927-1945), as I described in my initial review, reports the results of linguistic analyses (using word count) of Nazi propaganda concerning Jewish persons between 1927 and 1945 (stratified by month), before and after a cut-point date for the onset of the Holocaust, and in comparison to contemporaneous Nazi propaganda that was not in reference to Jewish persons. The focus was on two mentation constructs: agency and experience. Although other constructs—negative emotions, affiliation, health, purity, and death—were also examined.

As I originally opined, the topic struck as interesting, I did not identify any fatal flaws, and the authors seemed careful to qualify their interpretations in light study limitations.

However, I offered a few overall organizational comments and recommendations in reference to reported exploratory analyses, in addition to several smaller-scale comments and recommendations organized by manuscript section. I note the extent to which the authors’ addressed by various comments below, with issues for potential follow-up marked with three asterisks.

Organizational issues

A. “Agency and experience are distinct dimensions of mind but are 60 often moderately correlated (in our data, they were correlated at r = .32, p = .014).” I suggest using another reference to support this claim, moving your result to the Results section, and including the n and confidence interval.

The authors added a reference for the quoted statement, moved the correlation results to the Results, and reported the n and confidence interval for the analysis.

B. Relatedly, later in the manuscript, the exploratory “Threat Dictionary” analyses come a bit out of nowhere. I suggest restructuring so as to report these results in more detail in the Results section.

The authors now describe the Threat Dictionary analyses in the Method and report the results in full in the Results.

C. To this end, while the Introduction indicates that the focus is on two mind constructs, and the Discussion notes other non-mind constructs could be examined, other such constructs were in fact examined here (“Additional linguistic categories”). Even though the latter analyses were exploratory, reorganization and better clarification upfront of the range of constructs investigated is needed.

The authors now discuss the additional linguistic constructs in more detail in the Introduction and Method.

Introduction

The Introduction, while cursory, was nonetheless straightforward. Such that I felt like I was given enough background context, as a non-expert in linguistic analyses and mass violence, to make my own sense of the results. Though others might expect a detailed description of relevant prior literature.

Consistent with my comments about organization, the exploratory constructs can be noted after the hypotheses.

The Introduction continues to read straightforwardly, though I might reorder some later subsections as follows: Onset of the Holocaust > A note on perpetrators > Hypotheses > Additional linguistic constructs. The later subsection might also be relabeled: Exploratory analyses of additional linguistic constructs.

Method

The sampling procedure was very particular to this project, and not one I’m competent to critique (i.e., sources of translated Nazi propaganda).

***While no response to this comment was called for, of note, the authors have sought to address questions about the generalizability of their data (though some of this is currently a subsection in the Method, whereas said subsection may well be a better organizational fit in the Discussion).

The sample size (approximately 57,000 words from 140 sources vs. another more than 36,000 words from 123 sources) and time span (58 months vs. 51 months) seems sufficiently sizable for statistical analysis and drawing certain logical inferences. Though making this more explicit would be useful (e.g., with references to methodological discussions of sample size for linguistic analyses using longitudinal between-within designs, or reporting power or precision analyses).

The authors have added a sentence, with references, to explicit refer to the sufficiency of their sample size for their analyses and attendant interpretations.

Not being an expert in linguistic analyses, I’m not well positioned to critique the specific linguistic-analysis methodology that was used. But the authors did seem reasonably forthcoming about limitations of their method and issues that emerged when diagnosing data before conducting primary analyses.

While no response to this comment was called for, of note, the authors have reported more detail about the source of their data.

Consistent with my comments about organization, operationalizing all constructs examined (including exploratorily), and setting forth all related analysis plans, in the Method is recommended.

***Operationalizations for constructs examined are still variously reported across the Introduction and Method. I continue to suggest ensuring that all constructs are set forth individually and operationalized in the Method (e.g., for additional linguistic constructs).

While I am not an expert in longitudinal data analyses, the statistical analyses nonetheless seemed reasonable and appropriate (and adequately explained).

***While no response to this comment was called for, the authors did justify the choice they made about analyzing the data monthly, vs. alternative possibilities, in their narrative response—and such could be incorporated in text.

Given the range of analyses, providing justification for foregoing correction for experiment-wise error would be worthwhile.

The authors added a sentence to the Method to justify their choice to not correct for risk of experiment-wise error.

Results

Consistent reporting of confidence intervals is recommended.

The authors have appended confidence intervals to all primary and exploratory results.

Significance-difference testing results (e.g., in S1) also ought to be accompanied by a measure of effect size. To this end, more attention throughout to characterizing the magnitude of observed effects would be helpful for readers in appraising your interpretations.

The authors appended an effect size for one of the analyses reported in S1, and added a sentence in text to the Results to further draw attention to the small size of many of the observed effects.

Discussion

Like the Introduction, the Discussion reads as cursory but straightforward, for better or worse.

While no response to this comment was necessarily called for, of note, the authors have made some additions and reorganizations of the Discussion so that it better aligns with the Introduction.

The start of the Discussion would be made stronger and clearer if explicitly tied back to hypotheses.

***In the second paragraph of the Discussion, the authors highlight results relevant to their hypotheses. But whether those results were or were not consistent with their hypotheses could be more explicitly stated.

Consistent with my comments about organization, once the exploratory “threat” analyses are reorganized, results for all exploratory constructs would be worth interpreting/discussing.

***The authors added a subsection to the Discussion to make clearer their interpretation of the results concerning most of the additional linguistic constructs. Though the Speculations on increasing agency (where the threat construct results are discussed) and Additional linguistic constructs (where the other exploratorily examined constructs are discussed) subsections might be fused in the interest of consistent organization of constructs.

Additional limitations worth noting: Inherent uncertainty about the comprehensiveness of the sampling source, as a component of sampling validity. The reliability of the data extraction is seemingly unknown. Risk for experiment-wise error was not corrected.

***The authors added a sentence to the Limitations subsection of the Discussion to note the critique about sampling validity. However, the critiques about the lack of reported inter-rater reliability results for data extracting and coding, and the potential for some of the statistically significant but small-sized results being spurious due to a lack of controlling for risk of experiment-wise error, can still be noted and responded to.

Thank you for the opportunity to review this revised manuscript.

7. PLOS authors have the option to publish the peer review history of their article (what does this mean?). If published, this will include your full peer review and any attached files.

Reviewer #1: No

Reviewer #2: No

Reviewer #3: No

Reviewer #4: No

---

## [Author Response · Author response to Decision Letter 1]

22 Aug 2022

Please note that we have also included the following response in the document titled "Responses to Reviewers" in which bolded section headings and formatting may make it easier to read. Below is the text from that response letter:

August 22, 2022

To whom it may concern:

My colleagues and I would first like to sincerely thank the editor and four anonymous reviewers for their continued consideration of our manuscript, entitled “Dehumanization and mass violence: A study of mental state language in Nazi propaganda (1927-1945)” (Manuscript ID: PONE-D-21-38261). 

We received gracious feedback from the reviewers on July 20, 2022. Although we were able to successfully incorporate many of their suggestions to improve our piece, it was brought to our attention that there were still remaining points from Reviewer 4 to address. In the following, I will respond to each remaining point raised by Reviewer 4, noting our revisions, in the order they appeared in the decision letter. We again thank all four reviewers and the action editor for the time and effort they have taken to thoughtfully consider this work. 

Restructuring Introduction. We have now restructured the subsection of our Introduction in light of Reviewer 4’s suggestion: Onset of the Holocaust > A note on perpetrators > Hypotheses > Additional linguistic constructs (pp. 4-7). We agree this allows for a cleaner narrative progression. We have also relabeled the final subsection to “Exploratory analyses of additional linguistic constructs” (line 172). 

Limitations of our Data. Reviewer 4 suggested we transition our comments about the limitations of our data (lines 336-344) from the Methods to Discussion section. However, we felt it important to highlight the limitations of our data before presenting the analyses and results, so readers can more accurately appraise and interpret said results. Indeed, in this section we explicitly “encourage readers to keep these limitations in mind when evaluating our results.” Therefore, we elected to keep this subsection in the Methods section and hope Reviewer 4 understands our perspective. 

Moving Operationalizations to Methods. In the initial revision of this manuscript, we inadvertently provided operational—rather than purely conceptual—definitions of the additional linguistic constructs in the Introduction, as opposed to Methods section. Therefore, we agree with Reviewer 4’s recommendation that we operationalize all constructs in the Methods. We conceptually introduce each of the additional linguistic constructs in the Introduction (see “Exploratory analyses of additional linguistic constructs” subsection on pp. 5-7) before proceeding to operationally define them in the Methods (see “Psycholinguistic tools for additional linguistic constructs” subsection of pp. 9-10). 

Justifying Unit of Analysis. Reviewer 4 suggested we incorporate our justification for analyzing the data monthly, vs. alternative possibilities, into the main text. We have now done this (lines 256-259).

Discussing Results as Consistent or Inconsistent with Hypotheses. Reviewer 4 asked us to be more explicit regarding whether the patterns in mental state terms we observed we consistent with our hypotheses or not. However, we were intentionally reluctant to do so because we considered this study as an attempt to adjudicate between two compelling accounts that have both received theoretical and empirical support—and framed the introduction as such. For instance, note that we never say we agreed with the moral disengagement or limited relevance perspectives. Rather, this is how we lay out the hypotheses: “If mind denial facilitates violence by undermining moral concern, we would expect a negative trend in mental state terms from 1927 to the onset of the Holocaust. Conversely, if mind denial is of limited relevance to the instigation of mass violence, we would expect no consistent trend in mental state terms preceding the Holocaust.” We feel the balanced tone we adopt in the Discussion is consistent with our framing in the Introduction. 

Fusing Subsections in Discussion. We are thankful for Reviewer 4’s suggestion to fuse the “Speculations on increasing agency” and “Additional linguistic constructs” subsections of our Discussion in the interest of consistent organization of constructs. We have now done this (lines 478-488).

Noting Additional Limitations. Reviewer 4 pointed out that we neglected to note two additional limitations they pointed out in their initial review. Namely, the lack of reported inter-rater reliability results for data extraction and the potential for some of the statistically significant but small-sized results being spurious due to a lack of controlling for risk of experiment-wise error. We have now noted both of these limitations (lines 557-558; 561-563). 

Concluding remarks. We again thank the editor and reviewers for their continued consideration of our manuscript. We did our best to integrate the remaining feedback we received, and would genuinely appreciate your reconsideration of what is hopefully an improved paper.

Sincerely, 

Alexander Landry

Doctoral Student, Organizational Behavior

Stanford University

---

## [Decision Letter · Decision Letter 2]

8 Sep 2022

Dehumanization and mass violence: A study of mental state language in Nazi propaganda (1927-1945)

PONE-D-21-38261R2

Dear Dr. Landry,

We’re pleased to inform you that your manuscript has been judged scientifically suitable for publication and will be formally accepted for publication once it meets all outstanding technical requirements.

Kind regards,

Rashid Mehmood, PhD

Academic Editor

PLOS ONE

Additional Editor Comments (optional):

Reviewers' comments:

Reviewer's Responses to Questions

**Comments to the Author**

1. If the authors have adequately addressed your comments raised in a previous round of review and you feel that this manuscript is now acceptable for publication, you may indicate that here to bypass the “Comments to the Author” section, enter your conflict of interest statement in the “Confidential to Editor” section, and submit your "Accept" recommendation.

Reviewer #4: All comments have been addressed

2. Is the manuscript technically sound, and do the data support the conclusions?

Reviewer #4: Yes

3. Has the statistical analysis been performed appropriately and rigorously? 

Reviewer #4: Yes

4. Have the authors made all data underlying the findings in their manuscript fully available?

Reviewer #4: Yes

5. Is the manuscript presented in an intelligible fashion and written in standard English?

Reviewer #4: Yes

6. Review Comments to the Author

Reviewer #4: PONE-D-21-38261_R2

This second revision of the manuscript, Dehumanization and mass violence: A study of mental state language in Nazi propaganda (1927-1945), sought to further address five comments I noted in my review of the first revision. As noted below, I believe the authors have satisfactorily addressed these comments. Thank you for the opportunity to review this second revision of the manuscript.

The Introduction continues to read straightforwardly, though I might reorder some later subsections as follows: Onset of the Holocaust > A note on perpetrators > Hypotheses > Additional linguistic constructs. The later subsection might also be relabeled: Exploratory analyses of additional linguistic constructs.

The authors have incorporated these reorganizational and relabeling recommendations for the Introduction.

***While no response to this comment was called for, of note, the authors have sought to address questions about the generalizability of their data (though some of this is currently a subsection in the Method, whereas said subsection may well be a better organizational fit in the Discussion).

The authors declined to make this reorganizational recommendation. While deferring to the editors, I believe that this matter is within the prerogative of the authors.

***Operationalizations for constructs examined are still variously reported across the Introduction and Method. I continue to suggest ensuring that all constructs are set forth individually and operationalized in the Method (e.g., for additional linguistic constructs).

The authors added example parentheticals to operationalize their variables in the Method.

***While no response to this comment was called for, the authors did justify the choice they made about analyzing the data monthly, vs. alternative possibilities, in their narrative response—and such could be incorporated in text.

The authors incorporated their rationale into the Method.

***The authors added a subsection to the Discussion to make clearer their interpretation of the results concerning most of the additional linguistic constructs. Though the Speculations on increasing agency (where the threat construct results are discussed) and Additional linguistic constructs (where the other exploratorily examined constructs are discussed) subsections might be fused in the interest of consistent organization of constructs.

The authorized have incorporated this reorganizational recommendation.

***The authors added a sentence to the Limitations subsection of the Discussion to note the critique about sampling validity. However, the critiques about the lack of reported inter-rater reliability results for data extracting and coding, and the potential for some of the statistically significant but small-sized results being spurious due to a lack of controlling for risk of experiment-wise error, can still be noted and responded to.

The authors have incorporated both these critiques into to Discussion.

7. PLOS authors have the option to publish the peer review history of their article (what does this mean?). If published, this will include your full peer review and any attached files.

Reviewer #4: No

---

## [Editor Report · Acceptance letter]

14 Oct 2022

PONE-D-21-38261R2 

Dehumanization and mass violence: A study of mental state language in Nazi propaganda (1927-1945) 

Dear Dr. Landry:

I'm pleased to inform you that your manuscript has been deemed suitable for publication in PLOS ONE. Congratulations! Your manuscript is now with our production department. 

Kind regards, 

on behalf of

Dr. Rashid Mehmood 

Academic Editor

PLOS ONE